# High Precision Sea Surface Temperature Prediction of Long Period and Large Area in the Indian Ocean Based on the Temporal Convolutional Network and Internet of Things

**DOI:** 10.3390/s22041636

**Published:** 2022-02-19

**Authors:** Tianying Sun, Yuan Feng, Chen Li, Xingzhi Zhang

**Affiliations:** 1College of Information Science and Engineering, Ocean University of China, Qingdao 266005, China; suntianying@stu.ouc.edu.cn (T.S.); lichen8668@stu.ouc.edu.cn (C.L.); 2Key Laboratory of Physical Oceanography, Institute for Advanced Ocean Studies, Ocean University of China, Qingdao 266005, China; zhangxingzhi@ouc.edu.cn

**Keywords:** sea surface temperature, temporal convolutional network, Indian Ocean, Internet of Things

## Abstract

Impacted by global warming, the global sea surface temperature (SST) has increased, exerting profound effects on local climate and marine ecosystems. So far, investigators have focused on the short-term forecast of a small or medium-sized area of the ocean. It is still an important challenge to obtain accurate large-scale and long-term SST predictions. In this study, we used the reanalysis data sets provided by the National Centers for Environmental Prediction based on the Internet of Things technology and temporal convolutional network (TCN) to predict the monthly SSTs of the Indian Ocean from 2014 to 2018. The results yielded two points: Firstly, the TCN model can accurately predict long-term SSTs. In this paper, we used the Pearson correlation coefficient (hereafter this will be abbreviated as “correlation”) to measure TCN model performance. The correlation coefficient between the predicted and true values was 88.23%. Secondly, compared with the CFSv2 model of the American National Oceanic and Atmospheric Administration (NOAA), the TCN model had a longer prediction time and produced better results. In short, TCN can accurately predict the long-term SST and provide a basis for studying large oceanic physical phenomena.

## 1. Introduction

Sea surface temperature (SST) has an important impact on the health of regional marine ecosystems [1], and its changing trend may lead to the growth, reproduction, and distribution of marine species. Long-term SST forecasts on large-scale waters are of great significance to oceanic physical phenomena and help climate monitoring and early warning systems for flood and drought risks. Subsea changes are likely to leave traces on the sea surface through changes in sea-surface height (SSH) [2], so research on SST is important in order to investigate the subsurface parameters. According to Khedouri (1983) [3] and Ali (2004) [4], sea surface parameters are correlated with subsea ones. Yan et al. (1992) [5] estimated subsea parameters based on the ocean surface information. X. Wu et al. (2012) [2] showed that in a subpolar basin, the estimation accuracy of the subsurface temperature anomaly (STA) was improved by 40% through monthly SST and SSH data. M. Han et al. (2019) [6] estimated the subsurface temperature through parameters such as SST. M. Han et al. used the Copernicus Marine Environment Monitoring Service data sets (sea surface temperature anomaly (SSTA)), sea surface height anomaly (SSHA), and sea-surface salinity anomaly as input parameters of a convolutional neural network (CNN) model to predict the subsurface temperatures of the subsurface layers (a total of 57 layers) of the Pacific Ocean. The mean square errors (MSEs) of the subsurface temperature in all of those subsurface layers in the Pacific Ocean in January, April, July, and October were 0.2659, 0.3129, 0.5318, and 0.5160, respectively. So, research on SST is very important in many ways.

So far, investigators have focused on the short-term forecast of a small- or medium-sized area of the ocean. For example, research usually selects data sets covering thousands of meters to 2000 km and forecasts them one week or one month in advance. The following research involved the prediction of sea surface temperature in a small range in the medium and short term. Based on the sea temperature data of the past 7 days, 20 days, and 50 days, J. Dong et al. (2018) [7] made SST predictions at 1 day, 7 days, and 30 days in advance, respectively, on a small area of water in the Bohai Sea using a combined fully convolutional long short-term memory (LSTM) and CNN (CFCC-LSTM) model, with average MSEs of 0.1466 °C, 0.2722 °C, and 0.7260 °C, respectively. Xiao et al. (2019) [8] established an LSTM model in the East China Sea using 36 years of spaceborne sea surface temperature data; the model is accurate for the daily prediction of the short-term and medium-term sea surface temperature field. L. Guan et al. (2020) [9] divided the entire China Sea and its adjacent area into 130 small regions using the self-organizing map algorithm, constructed an LSTM model for each region to predict its SST, and found that the root-mean-square error (RMSE) of the forecasts at 1 month in advance was 0.5 °C. In summary, previous studies have mostly used data sets to make short-term predictions of regional SST, in which the selected feature is rather simple. The prediction of sea temperature over large areas of water and a long period has been rarely investigated. Moreover, as the forecast period is extended, the accuracy of the existing methods decreases [9]. Compared with the above, this paper used a TCN model to accurately predict long-term SSTs of a large area in the Indian Ocean. “Long term” means five years prediction, and “large area” means predicting on large-scale sea waters (40–110° E, 25° N–25° S) almost covering the Indian Ocean. Also, in this study, the influences of various factors on SST were fully considered when choosing features.

Also, from the above-mentioned studies it is evident that with artificial intelligence widely used, [10,11,12,13,14], deep learning models are gaining importance in the prediction of marine environmental elements. The prediction of sea surface temperature is usually solved as a time series problem, usually using LSTM. But LSTM has two disadvantages: it cannot extract spatial features and is prone to gradient problems when there is too much data [15,16]. By contrast, the TCN model can extract features in both time and space and is not prone to gradient problems because of its structure.

This study focused on the Indian Ocean to make an extended long-term SST forecast. By using multisource, multimodal air–sea data, a temporal convolutional network (TCN)-based model was constructed to perform a 5-year SST forecast on large-scale sea waters (40–110° E, 25° N–25° S, with a spatial resolution of 1° × 1°), thus realizing the ultra-long-term SST prediction on a large sea area. At the same time, theories of ocean physics and DL were combined in this study. The ocean surface is affected by ocean circulation and turbulence [5]. The large-scale SST annual cycle in the eastern equatorial Pacific is largely controlled by the changing depth of the mixed layer every year, while the depth of the mixed layer is mainly determined by solar radiation and the competitive effects of wind forces [17,18]. Therefore, this study selected multifactor, multilevel data of the ocean surface, the subsea, and the atmosphere as the input features to give the TCN model prior knowledge of physical oceanography in the training process, thereby achieving the goal of improving SST prediction accuracy through the powerful data-mining capability of a DL model.

The study of physical oceanographic phenomena in large-scale sea waters requires processing massive amounts of detailed data. The method proposed in this study only used data sets with low spatial resolution and time granularity to make ultra-long-term predictions about a large area of sea water. The aim is to predict trends in large-scale long-period SSTs and to improve the prediction accuracy as much as possible. Therefore, this method can have many applications, such as sea surface detection [19,20] and eddy current recognition, and it can be applied in practice and can play a key role in the study of some large-scale physical oceanographic phenomena, such as El Niño and Indian Ocean Dipole phenomena.

## 2. Materials and Methods

This study focused on the Indian Ocean (30–135° E, 30° N–66.5° S), which is the third largest ocean in the world. It influences climate anomalies in its surrounding areas, including Central and South America, the southern tip of Africa, southeastern Australia, northeast Asia, and other regions [21,22,23,24]. This study used the reanalysis data sets provided by the National Centers for Environmental Prediction, with a spatial resolution of 1° × 1°, to perform quality control and normalization on the observation data of various sources (ground, ship, radiosonde, wind balloon, aircraft, satellite, etc.). These data sets are characterized by many elements, a wide range, and long time range.

This study used the monthly ocean–atmosphere data of the sea area (40–110° E, −25–25° N) in the Indian Ocean obtained at https://psl.noaa.gov/data/gridded/ (accessed on 29 December 2021). The sea temperature forecast problem can be viewed as a time-series regression problem. Using monthly data is easier to study some ultra long-term marine phenomena, such as IOD and ENSO. The cycle of ENSO is long, usually 6–8 years, and using daily data is unrealistic.

The monthly data from 10 consecutive years were selected to predict the SST in the next 5 years. Table 1 shows the factors (a total of 81) that likely affect the SST of a given data point, and their monthly data were collected and divided into three categories: atmospheric parameters, sea-surface parameters, and subsea parameters. A total of 24 atmospheric parameters at different heights (1000, 850, 500, and 300 hPa), i.e., the data of atmospheric temperature (T), geopotential height (GPH), vertical velocity (W), relative humidity (RH), east–west wind speed (U), and north–south wind speed (V), were chosen. A total of 17 sea-surface parameters were included, i.e., the SSH of the centre point, the SST of the centre point, and the SSTs of 15 surrounding points (i.e., sub1_SST, sub2_SST, …, sub15_SST in the Table 1). A total of 40 subsea parameters were included, i.e., temperature, east–west ocean current (u), north–south ocean current (v), and sea-surface salinity (SSS) at each sea depth of the centre point (5, 15, 25, 35, 45, 55, 65, 75, 85, 95 m). The 1980–2008 data were used for a training dataset. The time span of features was 10 years. The training set was constructed by the sliding-window method, i.e., the sliding window slid backward 1 year at a time. The test set included the monthly atmospheric data of the ocean from 2004 to 2013 and their corresponding monthly SSTs from 2014 to 2018, making a total of 2533 pieces of data.

The TCN was adopted to perform deep learning (DL) and modelling with big data of ocean parameters. The TCN architecture adopted in this study was proposed by S. Bai (2018) [25]. TCNs use various ideas, including residual connection, dilated convolution, and causal convolution, to make them more effective in processing long-time-series/space problems. Of them, the residual connection (residual block) can to a certain extent eliminate the effects of vanishing gradients and gradient explosion that mar some DL networks [26]. By restricting the sliding direction of the convolution window, the causal convolution allows the prediction at time t (Yt) to be judged only through the input of x1 to xt − 1 before time t. Dilated convolution was originally adopted in the field of image segmentation, in which it decreases the loss of information and increases the receptive field while maintaining the same dimension numbers for both inputs and outputs. The formula for calculating the number of convolution kernels after dilated convolution is shown in Equation (1), in which K is the number of convolution kernels after dilation, k is the number of original convolution kernels, and d is the dilation rate of the neural network layer:K = (k − 1) × d + 1(1)

This study transformed the two-dimensional training set into a three-dimensional matrix (m, 120, 81) to input into the model. Each sample was a matrix (120, 81), one time step was 120, and each time step had 81 features. The layer number of the three-dimensional matrix was m, which equaled the number of samples. The numbers of convolution kernels were 8 and 24, dilations = (1, 2, 4, 8, 16, 32, 64, 128, 256), and stack = 1. A preliminary experiment showed that when the number of convolution kernels was set to 8, the model showed optimal efficiency, indicating that when training the model, at least the information of the past 8 months needed to be considered, and as the dilation factor increased, the historical time to be considered should also increase. The number of convolution kernels determines the number of feature maps generated in the convolution. The feature map contains information extracted from the output of the previous layer, and the information extracted through different convolution kernels differs. When the number of convolution kernels is too large, some occasional small data disturbances in the training set could be learned by the model, which could affect the accuracy of the model. When the number of convolution kernels is too small, the model’s ability to learn features becomes weak. In this study, we added more historical data to the output information by setting the dilation list, and the model was trained using the ordinary convolution method under the dilation rate of 1. The seasonal and interannual variation characteristics of SST in large sea areas required that more historical data be considered when predicting future sea temperatures. A comparison of the performance of ordinary and dilated convolutions on the data set of this study found that dilated convolution was more suitable for our purposes.

Figure 1 shows the operation of a sample in the TCN structure. A one-dimensional convolution (Conv1d) is performed on the samples of the input layer to obtain a matrix of size (120, 1, 24), which enters the first residual block for the operation (dashed box). The blue is the tensor of the calculation process of the TCN model. The red is the prediction results of TCN, which represents the sea surface temperature in the next 60 months. The input matrix of the residual block is convolved once in one dimension, and the same input matrix is convolved twice in one dimension, and the results are summed and passed to the second residual block, and the results obtained from both convolutions of the residual block are kept until the end for summing. Finally, the prediction result (60, 1) is mapped by the fully connected layer. The TCN model in this study was set up with nine residual blocks.

## 3. Results

The prediction model for the period from 2014 to 2018 was used as an example. The training set of the input model was a matrix of (120, 81, 50,660), of which 20% was used as the validation set to adjust the model parameters to obtain the optimal model. The model predicted the 5-year average monthly SSTs (60) from valid data points (2533) on the Indian Ocean basin and output a 60 × 2533 matrix. Various indicators, such as correlation, RMSE, accuracy (ACC), and the performances of the monthly model-predicted results against the observed results, were used to assess the accuracy of the TCN model in predicting the monthly average SST in the period from 2014 to 2018. By calculating the correlation between the actual 5-year SST time series of each data point and its corresponding model-estimated SST time series, the correlation distribution map was plotted. The higher the correlation, the more similar the trend of SST series, and the better the model fit. ACC represents the average prediction accuracy of the model on the data points. RMSE represents the magnitude of the error in predicting temperature, while ACC represents the average prediction accuracy of the model over the data points. According to the formula, ACC and RMSE are opposite trends.
(2)$$\mathrm{RMSE}=\sqrt{\frac{\sum_{i=1}^{m}{\left(X_{pred,i}-X_{real,i}\right)}^{2}}{m}}$$
(3)$$\mathrm{Accuracy}=1-\frac{\sum_{i=1}^{m}\left(\frac{\left|X_{pred,i}-X_{real,i}\right|}{X_{real,i}}\right)}{m}\;$$

### 3.1. Comparison of RMSE and ACC

The mean of the ACC value was 0.985, and the mean RMSE was 0.506 °C. As shown in Figure 2. The ACC values from August 2015 to June 2016 are all lower than the average ACC value, showing that the error increased in this period. The increase in error over this time period may be related to the occurrence of the IOD phenomenon in the Indian Ocean in 2015 and 2016, which manifested itself as an anomalous change in SST. In the other years, the errors are stable between 0.3 °C–0.5 °C. The results show that the average prediction error of the TCN model is relatively small, and the period from 2017 to 2018 is the late period of the prediction time and the error is stable, which indicates that the model is stable. It is further illustrated that the TCN model can accurately predict long-term SSTs.

### 3.2. Correlation

#### 3.2.1. Single Points Analysis

As shown in Figure 3, data points were densely populated in the correlation range of 0.90 to 0.97, while the correlation ranges below 75% all had fewer than 50 data points (i.e., only a few data points had a correlation of below 0.75). The correlations between the observed SST series and the model-estimated SST series at the 2533 valid data points in the Indian Ocean were all significantly positive and greater than 55%, and the data points in the correlation range of 55–65% only accounted for 1.54% of the total points (39/2533), while approximately half of the data points were in the correlation range of 85–95%. The correlations of the data points were mostly above 75%, with an average correlation of 88.23%. Therefore, the model’s fit to the data points was generally high. Figure 3 shows more clearly in which range the data point correlation is concentrated.

One data point from each of five correlation ranges was randomly selected, and the observed SST series curves and the predicted SST series curves of the five data points are shown in Figure 4. The closer the two curves, the higher the correlation. As shown in Figure 4a,b, the trend of the model-predicted SST series fit well with that of the observed SST series, so the two had a high correlation. The change of temperature data has a periodical feature, but the SSTs of the data points rose abnormally in 2016, observing the red line in Figure 4d,e. In 2015 and 2016, extreme positive IOD events and extreme negative IOD events occurred in the Indian Ocean, respectively. The points of the d and e parts of Figure 4 are located in the eastern region of IOD, so the prediction does not fit the real value well. So, in Figure 4d,e, due to the abnormal SST changes in 2016 at the data points, the deviation of the predicted SSTs from the observed SSTs is high, resulting in a low correlation between these data points. From these five randomly selected points, the temperature variation shows a cyclic change, one cycle per year, but each point is different. Within a cycle, some points have only one peak and trough, and some points have multiple peaks and troughs, and the present model has a good fit for all these points that show different trends. However, the fit is not accurate due to some anomalous changes in some points in some years. In conclusion, it can be seen from the plot of one point that the TCN model is able to basically fit the long-term sea surface temperature variation and is able to learn the periodic variation.

#### 3.2.2. Overall Spatial Analysis

First, in the analysis of TCN, Figure 5a shows the spatial distribution of the correlations of the predictions made by the TCN model. The data points in the correlation range of 55–65% were concentrated at the sea areas between −7° S and −2° S, 75° E and 100° E. The data points located in the middle of the studied waters were in the correlation range of 66–75%. In the low-latitude sea areas from −10° S to −25° S on the east coast of the Indian Ocean, the correlation was low, in the range of 75–85%. The data points in the coastal areas between the Somali Peninsula, the Arabian Peninsula, and the Indian subcontinent, as well as the seas in the west, especially the southwest, had a high correlation (over 85%). Overall, the fit to the SST time series on the Indian Ocean basin was above 75%, with an average correlation of 88.23%, indicating that the proposed model was very robust. According to Figure 5, the correlation degree of the region near the equator decreases. According to Figure 4, the data points d and e near the equator had large errors from June 2015 to June 2016. And the monthly RMSE began to rise from the summer of 2015 and gradually decreased after 2016.

Then there is a comparison between CNN and TCN, as shown in Figure 5a,b. Table 2 shows the RMSE, ACC and correlation of the CNN model and TCN model for comparison. As shown in Table 2, the RMSE, ACC and average correlation of the TCN model are higher than those of the CNN model. According to the experimental records, the training time of the CNN model is 12 h and that of the TCN model is about 6 h. The TCN model predicts more accurately and has a faster calculation speed than CNN when facing long-term prediction questions.

Next is the comparison between the CFSv2 model and the TCN model for five years. Figure 6a shows the spatial distribution of the correlations of the prediction results and actual values in the period from 2014 to 2018 using the CFSv2 model, released in 2010 by the National Oceanic and Atmospheric Administration (NOAA). The CFSv2 model only makes predictions for a period of 9 months. So, this study selected seven CFSv2 prediction models and compared the predictions by the CSFv2 model for the sea area of −25.039° S to 25.039° N, 40.312–109.687° E (with a resolution of 0.9375° × 0.9375°) with their corresponding actual values (with a resolution of 1° × 1°). The data points at the longitudes of 54.375°, 71.250°, 88.125°, and 105° E and the latitudes of 16.535° N, −0.472° S, and −16.535° S were excluded to ensure that most of the NOAA data points were close to the actual ones in terms of longitude and latitude. The results of the seven models were spliced, and then the five-year correlation was calculated. The average correlation of the CFSv2 prediction model was 87.27%, while that of the TCN model was 88.23%. The correlation of the CFSv2 model was lower than 0.55 in some data points, while that of the proposed model was always above 0.55. Figure 6b also shows that the correlation of the proposed model was generally higher than that of the CFSv2 model for data points north of the equator, indicating that the proposed model yielded high correlations and had a good ability to predict the changing trend of the SST long-time series. This shows that the prediction time of TCN model is longer and that the combined effect of the CFSv2 medium-term prediction model is not as good as the TCN model. Moreover, multiple CFSv2 models are needed to perform a long-term forecast, while one TCN model can make a 5-year forecast and thus is more functional than the CFSv2 model. The CFSv2 model can only predict the sea surface temperature in the next nine months.

Finally, in order to show the effect of the model more clearly, we collected the results of the first nine months of five years and compared them with the results of a CFSv2 model. From this point of view, we proved once again that TCN has strong long-term prediction ability, as shown in Figure 7a. The average correlation of the CFSv2 model is 89.8%, while that of TCN model is 96.1%, as shown in Table 3. It can be seen from Figure 7c that the correlation degree of the TCN model for most regions is greater than that of CFSv2 model. 

## 4. Discussion

This study made a long-term SST prediction on large-scale sea waters, in which the oceanic and atmospheric observation data of the past 10 years were used to forecast the SST in the next 5 years, and the prediction model was evaluated through the correlations of the predicted SST time series, RMSE, and ACC. The predicted SSTs in the period from 2014 to 2018 showed an average monthly error of 0.506 °C. The most important fact in this paper is that the 5-year correlation reached 88.23%

An advantage of this paper is the use of low spatial resolution and relatively small data sets, which yielded a high accuracy rate on the SST forecast on large sea areas over a long time scale. Generally, the number of deep learning samples is positively correlated with the prediction accuracy; this means that for data with higher resolution, the number of samples would also be more and the accuracy would be better. Current studies have focused on the SST forecast using a high-resolution dataset [6,7,8,9]. For example, Lei G. (2020) [9] et al. used a 0.05° × 0.05° resolution dataset forecasting 12 months in advance and RMSE was 0.66 °C. This study made a 5-year SST forecast of the entire Indian Ocean basin using 1° × 1° resolution dataset. But we achieved better results on RMSE (average monthly error of 0.506 °C).

Another strength of this paper is that although the forecast was long term, the error was stable, as shown in Figure 2. Generally speaking, the error increased with the increase of model prediction time. Lei G. [9] found that the accuracy of prediction one month in advance was the highest, with an error of 0.5 °C; When predicting 2 months in advance, RMSE increased to 0.59 °C; When predicting 12 months in advance, RMSE reached a peak of about 0.66 °C. So, it is obvious that with the increase of lead time, RMSE increases slowly. But the TCN model has a stable error, as shown in Figure 2a. In 2015 and 2016, the prediction error increased, while the error in other years was always small, ranging from 0.3 °C–0.5 °C. Although Indian Ocean Dipole occurred in the Indian Ocean in 2015 and 2016 [27], and this may have affected the error of the TCN model, it performed better than others. 

Another strength of this paper is that the study researches large-scale ocean physical phenomena. The study of marine physical phenomena usually involves analyzing the characteristics of seasonal and interannual changes over a medium or large area of the ocean considering multiple ocean–atmosphere factors. The fit of the SST of the entire ocean basin to the long-term prediction in this study was good, indicating that the proposed model is more suitable for the study of large-scale and long-term ocean physical phenomena.

Another advantage of this paper is that we selected many features. Large-scale ocean phenomena are affected by multiple factors, such as ocean current, sea wind, and sunlight. Model features in previous studies only included time-varying parameters of the sea surface (e.g., SST, SSS), but this study combined some theories of ocean physics with feature selection engineering to model multisource and multimodal data. These features included SST, sea subsurface temperature, flow velocity, atmospheric factors, and other factors to enable the model to learn the relationships between multiple features and SST, which is more in line with the requirements of studying physical phenomena.

The current studies on SST have focused on the forecasting of short and medium time scales in small sea areas, and the prediction accuracy decreases with increased forecasting time. Medium and short-term prediction generally refers to the time span of the next few days or months, and long-term prediction refers to the time span of more than several years. Among the models used in the past, CNNs cannot use the correlations between different feature time series (before and after), and LSTM models cannot extract the spatial dependency. TCNs can extract spatiotemporal information simultaneously and are thus superior to CNNs and LSTM models in many aspects. Since numerical values instead of images are inputted, TCNs have a faster calculation speed than CNNs while being less prone to the problem of gradient explosion. This study proposed an ultra-long-term SST prediction technology for large-scale sea waters based on multisource and multimodal observation data, which is conducive to using DL to develop deep-sea remote sensing technology. Limited by the data density and time span, the proposed model was only used to make 5-year SST predictions. We believe that by expanding the datasets, it will be possible to improve the prediction accuracy and extend the prediction period.

In addition, another strength of this paper is that TCN can extract its temporal and spatial characteristics. And when learning spatial location information, convolutional networks such as CNN are generally used, and the information is input into the model in the form of pictures, which requires a high computational workload. In the TCN model used in this study, only numerical values are input, making the calculation simpler and faster. Given the complexity of marine data systems, the model proposed in this study is apparently more suitable for marine data processing. The proposed model can output a 5-year SST time series at one time, making it more functional than the NOAA’s CFSv2 model, which can only make predictions 9 months in advance.

Due to the limited period of available data, this study only made a 5-year forecast. In future work, it should be possible to make SST forecasts for a longer period by increasing the data volume. As shown in Figure 6, in certain areas of the Indian Ocean (−10° S to 10° N, 40–110° E), the correlation was rather low, and Figure 4d,e shows clearly the research limitations. So special phenomenon (i.e., IOD) can not be observed. These areas showed a very strong nonlinear dynamic process, which needs simplification so that the prediction accuracy of DL algorithms in these areas can be improved. In future work, we will focus on improving the model’s prediction accuracy when a special phenomenon happens.

Regarding the reduced prediction accuracy of the TCN model near the equator in the inner ocean, we believe that the relationship between the convergence and divergence of ocean currents and their resulting upwelling and downwelling is nonlinear, and this nonlinear problem cannot be solved well by deep learning models. This nonlinear relationship is reflected in our model in that the variation of underwater UV and the variation of SST are nonlinear, thus causing a decrease in prediction accuracy near the equator. We also plan to conduct further research on this problem, such as designing appropriate linearization equations for specific situations and then adding them to deep learning to combine human knowledge with deep learning to achieve more accurate predictions.

In Figure 8 we have selected the region 70° E–90° E, equator, to 10° S to plot six plots of the real sea surface temperature. The TCN model predicted sea surface temperature for June, September, and December 2015. With the RMSE line graphs, we can see that the model prediction accuracy decreases in the second half of 2015, which affects the average correlation. So, we focus our discussion on the second half of 2015. Our prediction of the gradient of temperature is still accurate, especially in June, as can be seen in the comparison chart (south of the equator, gradually decreasing). The TCN model predicts this change in temperature. The same is true in September. This study focuses on the prediction of correlation, that is, the degree of fit between the true SST variation and the predicted SST variation. This study lays the foundation for large physical ocean phenomena, such as the IOD phenomenon, which manifests itself as the difference between the mean values of temperature in the east and west Indian seas. If we have a high level of correlation prediction of sea surface temperature, it will be of great help for such physical ocean phenomena, such as IOD, which are related to changes in SST.

## 5. Conclusions

In this study, the TCN model was used to predict the sea surface temperature of the Indian Ocean in the next five years using multisource and multimodal data in the past 10 years. The dataset used was from 1980 to 2008 for predicting monthly SSTs from 2014 to 2018. This study was about large ocean areas and long-term prediction. From 2014 to 2018, the RMSE of the proposed model was 0.420 °C, 0.556 °C, 0.650 °C, 0.487 °C and 0.417 °C, respectively. The 5-year SST’s average RMSE was 0.506 °C, and the correlation between the predicted SST time series and the observed SST time series was 88.23%. Through the results analysis, the TCN method was stable and reliable, with high accuracy. The model contributes to the long-term prediction of SST, thus greatly aiding the study of physical phenomena in the ocean. The model performed better in ordinary years than in years with abnormal IOD events. In the future, we will add other data to improve the model’s prediction accuracy for abnormal years.

In sum, since long-term prediction requires a lot of data, the traditional LSTM model does not work, which is a problem for long-term prediction. The TCN model is suitable for long-term forecasting and has relatively high accuracy. Secondly, we found that using multi-elements is good for improving long-term prediction, because long-term SST change is about the long-term sea–air dynamics process, and multi-element learning is more consistent with this. In the results, it was also proved that our multi-element learning can improve prediction for a long-term period. Finally, we focused on the correlation to measure the effect of the model. This is beneficial for large-scale oceanic phenomena. IOD is the temperature difference between regions, and it would be one-sided to focus only on the average error of all points of the RMSE. Both regions have errors

## Figures and Tables

**Figure 1 sensors-22-01636-f001:**
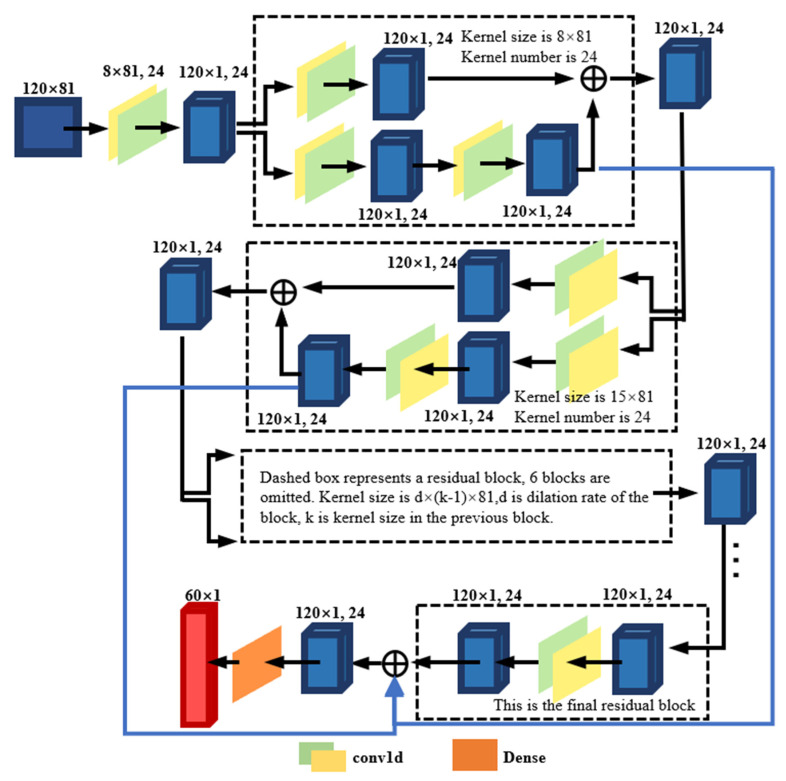
TCN structure, including nine residual blocks and one fully connected layer.

**Figure 2 sensors-22-01636-f002:**
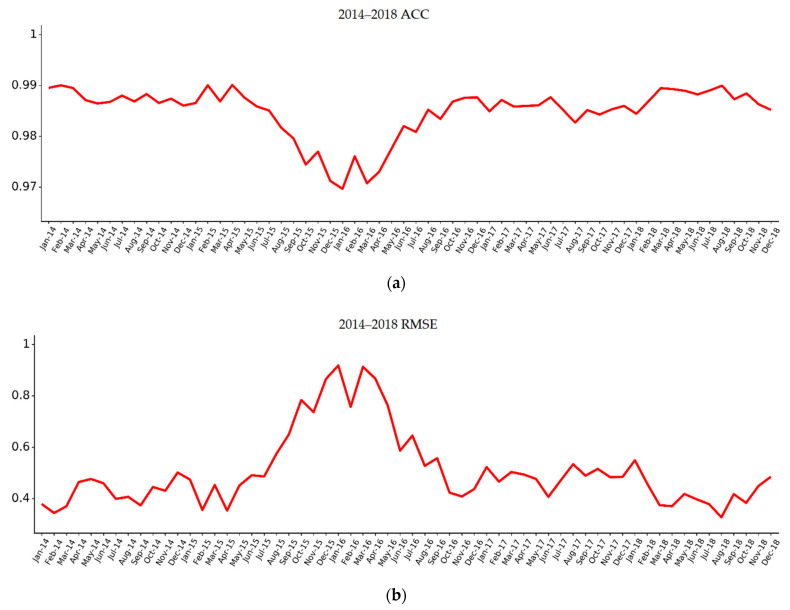
(**a**) The ACC histogram for each month from 2014 to 2018; (**b**) the RMSE line chart for each month from 2014 to 2018.

**Figure 3 sensors-22-01636-f003:**
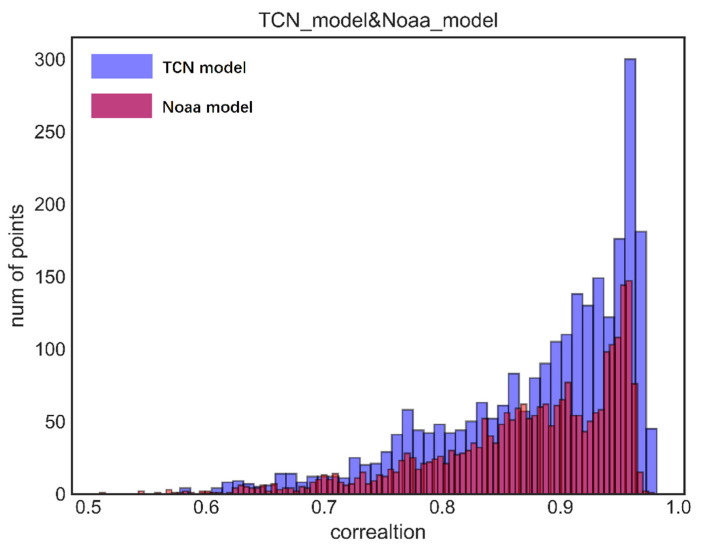
Histogram of the correlation distribution of the TCN and NOAA prediction model for the period from 2014 to 2018.

**Figure 4 sensors-22-01636-f004:**
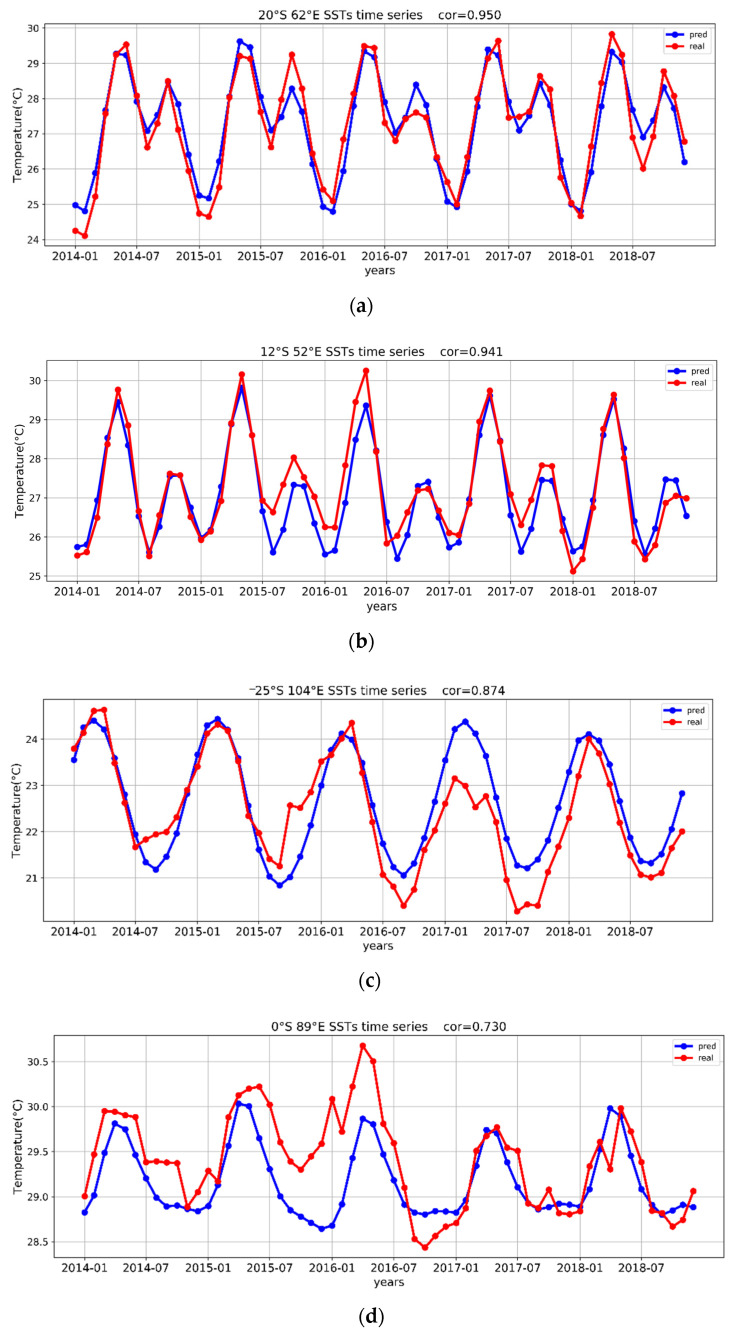
The changes in the observed SSTs and the predicted SSTs in the period from 2014 to 2018 are reflected by five data points that were randomly selected from five correlation ranges. (**a**–**e**) take the five years SST curve of points with correlation of 60–100%.

**Figure 5 sensors-22-01636-f005:**
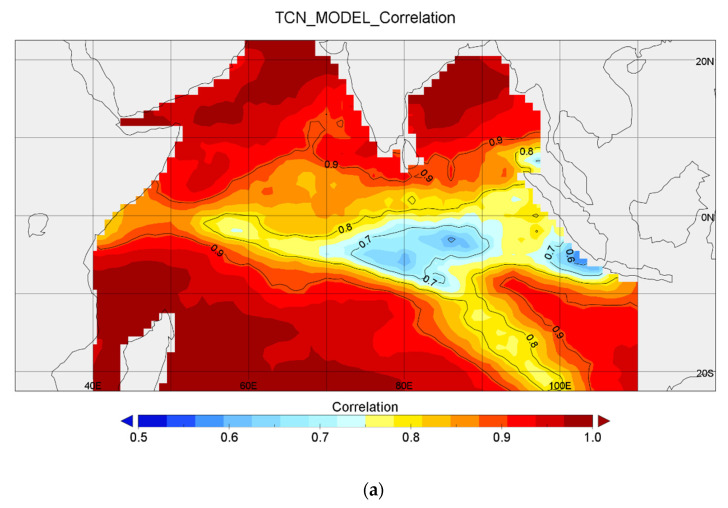
(**a**) Correlation between the TCN-predicted SST time series and the actual SST time series. (**b**) Correlation between the CNN model-predicted SST time series and the actual SST time series.

**Figure 6 sensors-22-01636-f006:**
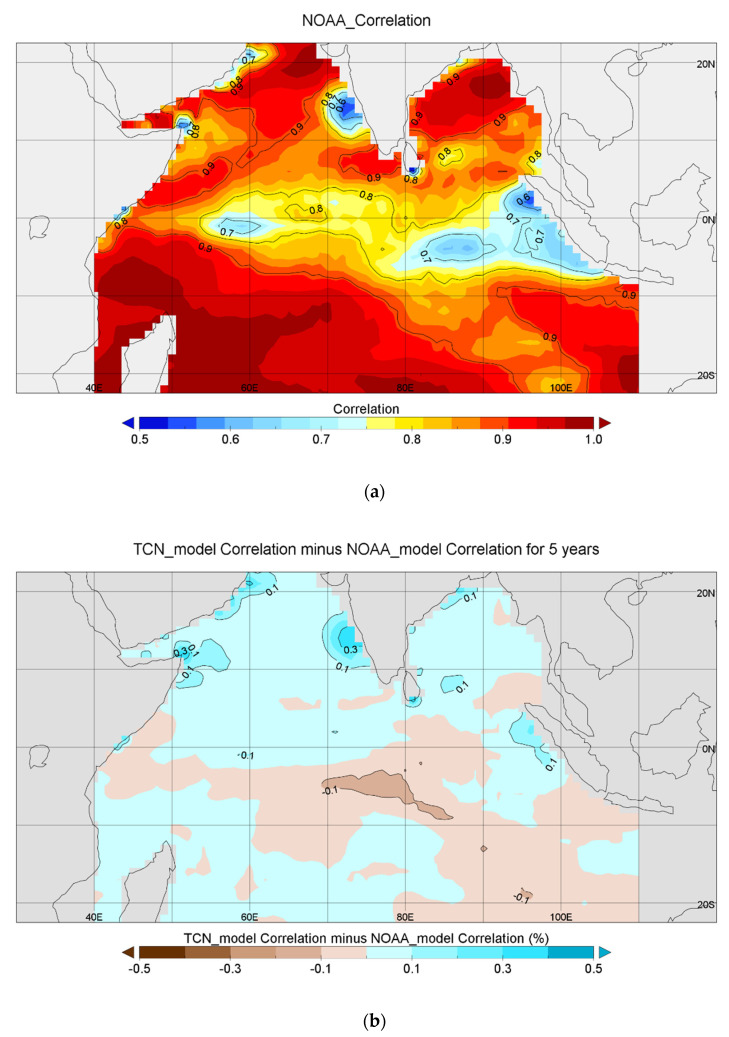
(**a**) Correlation between the Climate Forecast System version 2 (CFSv2) model-predicted SST time series and the actual SST time series. (**b**) The correlation of each point of TCN model minus the correlation of the corresponding points of cfsv2.

**Figure 7 sensors-22-01636-f007:**
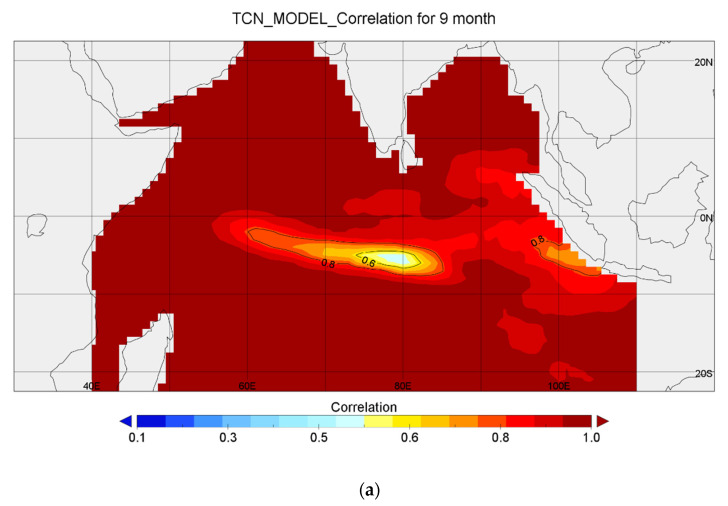
(**a**) The correlation distribution of the TCN model from January 2014 to September 2014. (**b**) The correlation distribution of the CFSv2 model from January 2014 to September 2014. (**c**) The correlation distribution of the TCN model minus the CFSv2 model from January 2014 to September 2014.

**Figure 8 sensors-22-01636-f008:**
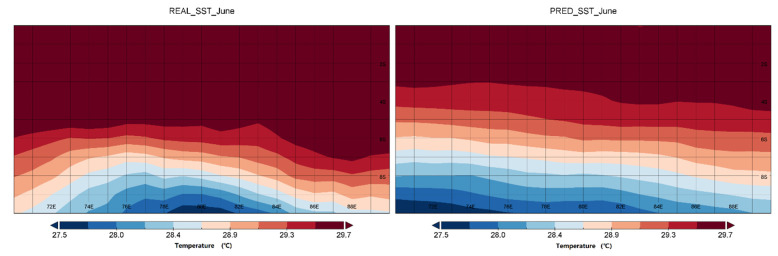
Comparison of regional real and predicted sea surface temperature (70° E–90° E, equator to 10° S) for June, September and December 2015.

**Table 1 sensors-22-01636-t001:** 81 Factors used in the model.

Parameters of Atmosphere					
AT 1000 hpa	AT 850 hpa	AT 500 hpa	AT 300 hpa	GPH 1000 hpa	GPH 850 hpa
GPH 600 hpa	GPH 400 hpa	VV 1000 hpa	VV 850 hpa	VV 600 hpa	VV 400 hpa
RH1000 hpa	RH 850 hpa	RH 600 hpa	RH 400 hpa	ZWS 1000 hpa	ZWS 850 hpa
ZWS 600 hpa	ZWS 400 hpa	MWS 1000 hpa	MWS 850 hpa	MWS 600 hpa	MWS 400 hpa
**Parameters of Subsea**					
SS 5 m	SS 15 m	SS 25 m	SS 35 m	SS 45 m	SS 55 m
SS 65 m	SS 75 m	SS 85 m	SS 95 m	u 5 m	u 15 m
u 25 m	u 35 m	u 45 m	u 55 m	u 65 m	u 75 m
u 85 m	u 95 m	v 5 m	v 15 m	v 25 m	v 35 m
v 45 m	v 55 m	v 65 m	v 75 m	v 85 m	v 95 m
ST 5 m	ST 15 m	ST 25 m	ST 35 m	ST 45 m	ST 55 m
ST 65 m	ST 75 m	ST 85 m	ST 95 m		
**Parameters of Sea Surface**					
SST	sub1_SST	sub2_SST	sub3_SST	sub4_SST	sub5_SST
sub6_SST	sub7_SST	sub8_SST	sub9_SST	sub10_SST	sub11_SST
sub12_SST	sub13_SST	sub14_SST	sub15_SST	SSH	

**Table 2 sensors-22-01636-t002:** Comparison of RMSE and ACC for CNN and TCN models.

	CNN	TCN
RMSE	0.529	0.506
ACC	98.4%	98.5%
CORRELATION	84.7%	88.23%

**Table 3 sensors-22-01636-t003:** Comparison of correlation for TCN model and CfSv2 model.

for Nine Months	TCN	CFSv2
CORRELATION	96.1%	89.8%

## Data Availability

Omega: https://psl.noaa.gov/cgi-bin/db_search/DBListFiles.pl?did=198&tid=96029&vid=664. Air temperature: https://downloads.psl.noaa.gov/Datasets/ncep.reanalysis.derived/surface/air.mon.mean.nc. Uwind: https://downloads.psl.noaa.gov/Datasets/ncep.reanalysis.derived/surface/uwnd.mon.mean.nc. Vwind: https://downloads.psl.noaa.gov/Datasets/ncep.reanalysis.derived/surface/vwnd.mon.mean.nc. Geopotential height: https://psl.noaa.gov/cgi-bin/db_search/DBSearch.pl?Variable=Geopotetial+Height&group=0&submit=Search. Specific Humidity: ftp://ftp2.psl.noaa.gov/Datasets/ncep.reanalysis.derived/pressure/shum.mon.mean.nc. Potential temperature: https://psl.noaa.gov/cgi-bin/db_search/DBListFiles.pl?did=98&tid=91102&vid=1913. Salinity: https://psl.noaa.gov/cgi-bin/db_search/DBListFiles.pl?did=98&tid=91102&vid=1914. V current: https://psl.noaa.gov/cgi-bin/db_search/DBListFiles.pl?did=98&tid=91102&vid=1920. U current: https://psl.noaa.gov/cgi-bin/db_search/DBListFiles.pl?did=98&tid=91102&vid=1918. Sea surface height: https://psl.noaa.gov/cgi-bin/db_search/DBListFiles.pl?did=98&tid=91102&vid=1916. SST: ftp://ftp2.psl.noaa.gov/Datasets/COBE2/sst.mon.mean.nc.

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
