# Peer review of "High Precision Sea Surface Temperature Prediction of Long Period and Large Area in the Indian Ocean Based on the Temporal Convolutional Network and Internet of Things"

_sensors, 2022, doi:10.3390/s22041636_

Round 1

Reviewer 1 Report

1.The abstract must been modified. And it is better for the authors to explain more about the details of the numerical results.
2.The authors should further improve the writing and make the figures more clear. The figures in the paper are too small. It is better to enlarge the figures.

3.The DL-TCN model takes into account an amount of data. My question is about how the data taken at 65m depth reflect some at the surface. It is possible that they introduce some errors in estimations. This must be clarified.

4.If you are using the model for whole fields, you must update your references. But if you are using the model for single points, you may use a quite different strategy.

5.The authors didn’t provide a comparison between DL model and conventional physical-based method to prove the advantage and limitation of DL model. This is something that needs to be put emphasis on discussing.

6.After a careful reading I miss a more proper discussion relating the general map of currents with the results. I think a practical application is needed. So, a discussion of the results looking for the relationship with the general scheme of currents must be included.

Author Response

Dear reviewers:

      We have modified this article to meet your requirements and the green is new changed part. According to your suggestions, we make the following improvements:

  1. We have carefully discussed the key points of the paper and written many versions of abstracts. In the abstract of the current version, we only mentioned the results of correlation, not the experimental results of RMSE and ACC. And we think this is enough for the abstract. RMSE and ACC, as secondary parts, strengthen the demonstration of good model effect. This paper uses a relatively large space to explain the correlation degree, such as the correlation degree distribution map, the comparison of the correlation degree of the model, and the SST curve of the data points. So the summary is appropriate

  2. The missing title of the figure has been added

  1. Generally speaking,subsurface oceanic process including turbulent mixing , Kelvin wave and vertical entrainment could affect sea surface temperature directly or indirectly,it makes sense to take data at some depths like 65m .

  1. TCN model is applied on the whole SST field.

  1. The conventional methods, for example, artificial neural network(ANN), cannot obtain large-scale and long-term predictions. Our team has also done experiments to predict the long-term sea surface temperature by constructing ANN, but the results were very unsatisfactory, the loss of the model didn’t decrease, and the prediction error of sea surface temperature was very large.

Reviewer 2 Report

The authors did great work to predict the SST in the Indian Ocean. However, this manuscript must be improved since too many repetition statements, lack of references, views without clear supporting information (i.e., figure/table) are everywhere, and some sentences are hard to understand the meaning. Please read my comment in the attached pdf file for further details. 

Author Response

Thank you for your work. We have modified this article to meet your requirements and the green is new changed part. repetition statements have been deleted. lack of references and views without clear supporting information have been added. some understandable sentences have been  reinterpreted.
